# Exploring the Antibacterial Potential and Underlying Mechanisms of *Prunella vulgaris* L. on Methicillin-Resistant *Staphylococcus aureus*

**DOI:** 10.3390/foods13050660

**Published:** 2024-02-22

**Authors:** Ziyin Li, Qiqi He, Feifei Xu, Xinxin Yin, Zhuofan Guan, Jia Song, Zhini He, Xingfen Yang, Chen Situ

**Affiliations:** 1Food Safety and Health Research Center, NMPA Key Laboratory for Safety Evaluation of Cosmetics, Guangdong Provincial Key Laboratory of Tropical Disease Research, School of Public Health, Southern Medical University, Guangzhou 500515, China; 13751322925@163.com (Z.L.); fjykdxxff0924@126.com (F.X.); yinaym@foxmail.com (X.Y.); 13018578028@163.com (Z.G.); jia_song1991@126.com (J.S.); hezhinihzn@126.com (Z.H.); 2Institute for Global Food Security, School of Biological Sciences, Queen’s University Belfast, Belfast BT9 5DL, UK; qhe03@qub.ac.uk

**Keywords:** *Prunella vulgaris* L., MRSA, cell membrane, enzyme activity, bacterial metabolism

## Abstract

*Prunella vulgaris* L. (PV) is a widely distributed plant species, known for its versatile applications in both traditional and contemporary medicine, as well as in functional food development. Despite its broad-spectrum antimicrobial utility, the specific mechanism of antibacterial action remains elusive. To fill this knowledge gap, the present study investigated the antibacterial properties of PV extracts against methicillin-resistant *Staphylococcus aureus* (MRSA) and assessed their mechanistic impact on bacterial cells and cellular functions. The aqueous extract of PV demonstrated greater anti-MRSA activity compared to the ethanolic and methanolic extracts. UPLC-ESI-MS/MS tentatively identified 28 phytochemical components in the aqueous extract of PV. Exposure to an aqueous extract at ½ MIC and MIC for 5 h resulted in a significant release of intracellular nucleic acid (up to 6-fold) and protein (up to 10-fold) into the extracellular environment. Additionally, this treatment caused a notable decline in the activity of several crucial enzymes, including a 41.51% reduction in alkaline phosphatase (AKP), a 45.71% decrease in adenosine triphosphatase (ATPase), and a 48.99% drop in superoxide dismutase (SOD). Furthermore, there was a decrease of 24.17% at ½ MIC and 27.17% at MIC in tricarboxylic acid (TCA) cycle activity and energy transfer. Collectively, these findings indicate that the anti-MRSA properties of PV may stem from its ability to disrupt membrane and cell wall integrity, interfere with enzymatic activity, and impede bacterial cell metabolism and the transmission of information and energy that is essential for bacterial growth, ultimately resulting in bacterial apoptosis. The diverse range of characteristics exhibited by PV positions it as a promising antimicrobial agent with broad applications for enhancing health and improving food safety and quality.

## 1. Introduction

A significant global health and food safety concern persists in both developed and developing countries due to foodborne illnesses arising from the ingestion of microbial pathogens through food or drink [1]. It is estimated that contaminated food consumption leads to illness in one out of every ten individuals and causes 420,000 deaths annually worldwide [2]. The persistence of microorganisms in food, particularly bacteria, presents a significant concern, given their ability to proliferate and thrive under varying environmental conditions. This can result in both food spoilage and foodborne diseases in humans. The manifestation of foodborne illnesses is primarily due to the enterotoxins produced by pathogenic microorganisms following enzymatic breakdown in the gastrointestinal tract, which is resistant to heat treatment. It is important to highlight that staphylococcal food poisoning continues to be the most prevalent foodborne illness globally [3].

Concomitantly, the acquisition of antibiotic resistance by Staphylococci has been a significant driver of its evolution. The emergence of methicillin-resistant *Staphylococcus aureus* (MRSA), which is capable of breaking down virtually all ß-lactam drugs, represents a formidable public health challenge for the 21st century [4]. Discovered in 1961, epidemic strains of MRSA have swiftly disseminated and emerged as the predominant causative agent of bacteremia in the UK and Europe during the 1990s [5]. MRSA infections in healthcare settings are notorious for their high rates of morbidity and mortality worldwide. It has been designated as a global priority pathogen by the World Health Organization (WHO) and remains the most frequently reported pathogen at the global scale [6]. With the currently limited number of antimicrobial agents in development, there is a pressing need to investigate novel, alternative strategies to combat the MRSA threat [7].

Medicinal plants have been utilized as natural antibacterial agents in the food and pharmaceutical industries for centuries [8]. Plant-based preservatives are generally recognized as safe (GRAS), and therefore play a pivotal role in food preservation and functional development [9]. In light of this, exploring alternative strategies to minimize the use of antibiotics in agricultural food production while ensuring consumer satisfaction has merged as a paramount concern within the scientific community. Medicinal plants contain a diverse range of active phytochemicals, such as alkaloids, terpenoids, flavonoids, and phenolic compounds, which have been proven to possess a broad spectrum of antibacterial properties. Notably, these bioactive compounds are characterized by a low incidence of microbial resistance, making them invaluable in combating resistant bacterial infections. 

Belonging to the Labiate family, *Prunella vulgaris* L. (PV) is a highly valued, edible–medicinal plant that thrives across diverse geographical regions, including Korea, Japan, China, and Europe [10]. Rooted in Asian and European traditional medicine, this versatile herb has long been utilized to treat a large array of health conditions, from sore throats to burns and wounds [11]. Moreover, PV has gained widespread popularity in both food and beverages in China and Europe [12,13], with the iconic recipe of “Wang Lao Ji Liang Tea” serving as a health-promotion drink for over a century in southern China [14]. Notably, PV holds a prominent position as a standard medicinal material in Chinese pharmacopoeia, attesting to its enduring importance.

PV has garnered considerable attention due to its numerous activities and remarkable pharmacological properties, such as antioxidant, anticancer, anti-viral, anti-inflammatory, antibacterial, and neuroprotective effects [13,15], highlighting its immense potential for clinical use, particularly as an antimicrobial agent [16]. Studies have demonstrated that PV extracts exhibit antimicrobial activity against a range of disease and foodborne pathogens, including *Salmonella* Typhi, *Escherichia coli*, *S. aureus,* and *Klebsiella pneumonia* [17]. When employed as a preservative, a combination of PV aqueous–ethanol extract, *Hysoppus officinalis* and *Melissa officinalis,* resulted in the prolonged shelf-life of fish [18]. Additionally, the synergistic effects resulting from a combination of botanicals and antibiotics have been observed in multiple studies, especially in the context of traditional Chinese medicine (TCM) [19,20].

Previous studies have primarily focused on the therapeutic properties of isolated compounds, such as flavonoids and phenolic acids derived from PV [21]. However, given the intricate composition of the plant, a singular emphasis on the biological activity of one specific group of components may not provide a comprehensive understanding of its full functionality. Despite several investigations reporting varying levels of antibacterial activity of PV, the mode of action has yet to be thoroughly investigated. Hence, this present study sought to characterise the phytochemical composition relevant to the antibacterial properties of PV extract, comprehensively elucidating the underlying mechanisms of action against MRSA. 

## 2. Materials and Methods

### 2.1. Reagents and Chemicals

Dried aerial parts of PV were obtained from the Traditional Chinese Medicine Piece Factory of Guangdong Medicinal Material Company (Product serial number: X2419412. Foshan, China). Mueller–Hinton agar (MHA) (Catalog No. 0405B) and Mueller–Hinton broth (MHB) (Catalog No. CM0337B) were purchased from Thermo Scientific Oxoid™ (Basingstoke, UK). The AKP assay kit, ATPase Assay Kit, Total Superoxide Dismutase (T-SOD) Assay Kit, and Bicinchoninic Acid (BCA) Protein Quantification Kit were obtained from Abcam (Cambridge, UK). Penicillin, erythromycin, lysozyme, ethylenediamine tetraacetic acid, iodonitrotetrazolium chloride, phosphate buffer, and other chemicals were purchased from Sigma-Aldrich (Gillingham, UK). All chemicals used in this experiment were of analytical grade. 

### 2.2. Plant Materials and Extraction

PV was initially ground into a fine powder using a planetary ball mill PM 100 (Retsch, Hann, Germany) and subsequently extracted with water, 70% ethanol (70:30, *v*/*v*), and absolute methanol, respectively. For aqueous extraction, the powdered plant material was macerated with water at the ratio of 1:10 (*w*/*v*) for 2 h at room temperature (22 °C) with stirring, and then extracted by boiling the water to 100 °C for 2 h [22]. For organic extractions, the powdered plant samples were mixed with methanol or 70% ethanol for 24 h at room temperature [17]. After centrifuging the mixtures at 10,000× *g* for 40 min, the solvent was removed using a rotary evaporator (Buchi, Flawil, Switzerland) and dried using a Lablyo vacuum freeze dryer (Frozen in Time, York, UK). The extraction yields for aqueous, methanol, and 70% ethanol were recorded as 10.02%, 4.4%, and 6.25%, respectively.

### 2.3. UPLC-ESI-MS/MS Analysis of PV Extract

PV crude aqueous extract (1 mg) was thoroughly dissolved in 1 mL PBS and filtered using a 0.45 μm filter membrane for UPLC-MS/MS analysis, using a Kinetex C18 column (3.0 mm × 100 mm × 2.6 μm) (Phenomenex, Torrance, CA, USA) with a mobile phase consisting of 10 mM aqueous ammonium acetate for mobile phase A and methanol for mobile phase B. The elution was achieved through a linear gradient process: 0–10 min, B: 5%–95%; 10–11 min, B: 95%; 11–12 min, B: 95%-5%; 12–15 min, B: 5%-5%. The column temperature was maintained at 35 °C, while the mobile phase flowed at a rate of 0.4 mL/min and an injection volume of 5 µL. Using a Q-Executive high-resolution mass spectrometer (Thermo Fisher Scientific, San Jose, CA, USA) in negative-ion (ESI^−^) mode, the ESI-MSn experiment was carried out over a scan range of 100–1500 *m*/*z*. The ionization source conditions were set to a capillary voltage of 2.5 kV, source temperature of 130 °C and desolvation temperature of 350 °C, respectively. The resolution of first-order mass spectrometry was set to 70,000 (san range 65–800 *m*/*z*). The second-order mass spectrometry was executed with a resolution of 17,500 and an AGC target of 8 × 10³, while the TopN and collision energy were set at 5 and 10, 30, and 50, respectively. All the acquired data were initially processed using Compound Discoverer 3.0 (CD3.0, Thermo Fisher Scientific, Waltham, MA, USA), and further scrutinized and compared through mzCloud, mzVault and ChemSpider databases. The matching degree of 70% was designated as the criterion for the comparative analysis.

### 2.4. Bacterial Strain and Preparation of Inoculum

EMRSA-15 (NCTC13142), obtained from the Public Health England Culture Collections, was used throughout this study. The bacterial suspension was prepared by transferring 3–5 colonies from an overnight MH agar plate culture to a tube containing 5 mL of MH broth. The broth culture was allowed to incubate at 37 °C until it reached or exceeded the 0.5 McFarland standard in visible turbidity. Subsequently, it was diluted to 1:100 to serve as the inoculum [23]. 

### 2.5. Antibacterial Activity and Synergistic Testing of PV Extracts 

#### 2.5.1. Determination of Minimum Inhibitory Concentration (MIC) and Minimum Bactericide Concentration (MBC)

The microbroth dilution method recommended by the British Society for Antimicrobial Chemotherapy was used to determine the minimum inhibitory concentration (MIC) of PV extracts [24]. Serial double dilutions of the aqueous, methanol, and ethanol PV extracts were prepared in MH broth and transferred to a 96-well microtiter plate. Each testing plate included a growth control with organism and broth only, as well as a sterility control with media only. The bacterial suspension was then incubated with the extracts for 24 h at 37 °C. The MIC was defined as the lowest concentration of plant extract at which no visible growth was observed in the wells. The minimum bactericidal concentration (MBC) was measured by subculturing 20 μL of the wells with no visible bacterial growth for a further 24 h incubation. The MBC was defined as the lowest concentration of plant extract that prevented the growth of MRSA on the agar plate. The test was repeated three times under the same conditions. 

#### 2.5.2. Growth Kinetic Analysis

To explore the dynamic interplay between the plant extracts and the pathogenic organism, we employed a previously described method [25]. In a 96-well plate, 100 µL of MRSA suspension ($1\times 10^{6}$ CFU/mL), previously prepared as described, was inoculated and mixed with 100 µL of the PV aqueous extract at a final concentration of ½ MIC and 1 MIC. The bacterial growth was measured by the optical density at 600 nm at time zero, and then every hour for 24 h at 37 °C, using a FLUOstar Omega multi-mode microplate reader, (BMG Labtech, Ortenberg, Germany). This experiment was conducted three times under identical conditions.

#### 2.5.3. Synergistic Testing

The potential anti-MRSA activity of PV extracts combined with penicillin or erythromycin was quantified by the fractional inhibitory concentration index (FICI). checkerboard methods were employed to analyse the synergy between PV extracts and antibiotics [20]. MRSA were cultured in the presence of PV extracts, with final concentrations ranging from $\frac{1}{64}$ MIC to 2 MIC, in conjunction with either penicillin or erythromycin at concentrations ranging from $\frac{1}{256}$ MIC to 1 MIC. The fractional inhibitory concentration index (FICI) was used to calculate the potential synergy. The experiments were conducted similarly to the MIC determination, as described above.

The FIC index was calculated using following formula:(1)$${FIC}_{antibiotic}=\frac{{MIC}_{antibiotic\;in\;combintion}}{{MIC}_{antibiotic\;alone}}$$
(2)$${FIC}_{PV\;extract}=\frac{{MIC}_{PV\;extract\;in\;combintion}}{{MIC}_{PV\;extract\;alone}}$$
(3)$$FICI={FIC}_{antibiotic}+{FIC}_{PV\;extract}$$
where FICI ≤ 0.5 indicated synergy; 0.5 < FICI ≤ 1 indicated additivity; 1 < FICI < 2 indicated indifference; FICI ≥ 2 indicated antagonism.

### 2.6. Antibacterial Mechanism of PV against MRSA

#### 2.6.1. Integrity of Cell Membrane

To assess the impact of PV aqueous extract on the integrity of the cell membrane, nucleic acid leakage and the release of proteins into the bacterial cell culture were examined using modified methods described by Kamjumphol, et al. [26]. Bacteria were cultured in MHB until reaching the logarithmic phase, then mixed with the PV extract at final concentrations of 1/2 MIC and 1 MIC. After 5 h, the bacterial suspension was centrifuged at 10,000 rpm and 4 °C for 15 min. The supernatant was filtered through a 0.22 µm cellulose ester microporous membrane and the leakage of the bacterial cells was measured at 260 nm using a UV-vis spectrophotometer (Bio Tek, Winooski, VT, USA).

The protein release was measured following the protocol described by Fang et al. [25]. Bacteria were cultivated in fresh MHB until reaching the logarithmic phase, after which they were treated with the PV extracts at concentrations of 1/2 MIC and 1 MIC for 5 h. The resulting culture solution was then centrifuged at 12,000 rpm for 5 min, and the supernatant was mixed with 100 μL of reagent solution, containing a 1:50 mixture of Copper and BCA reagents. Assay buffer was used as the control. The optical density (OD) was measured at 562 nm, and the protein content of corresponding samples was determined using a standard curve.

#### 2.6.2. Integrity of Bacterial Cell Wall

Alkaline phosphatase (AKP) activity was measured to evaluate the impact of PV aqueous extract on cell wall integrity. Bacteria grown to the logarithmic phase were collected, centrifuged, and washed with 0.85% NaCl solution three times before being diluted to obtain an adjusted OD_600 nm_ = 2.0 suspension [27]. Next, 1 mL of bacterial suspension was mixed with PV extracts at final concentrations of 1/2 MIC and 1 MIC and incubated for 5 h at 37 °C. Bacterial cells were collected by centrifugation at 8000 rpm for 10 min and washed three times with PBS. The cellular AKP activity was measured using an AKP kit, with readings taken using a FLUOstar Omega microplate reader, (BMG Labtech, Ortenberg, Germany). PBS served as the control for the measurement.

#### 2.6.3. Impact on Cellular Adenosine Triphosphatase (ATPase) and Superoxide Dismutase (SOD)

The effects of PV aqueous extract on ATPase and SOD were investigated employing the ATPase Assay Kit and Total Superoxide Dismutase (T-SOD) Assay Kit, following the manufacturer’s instructions. The bacterial suspension was prepared as previously described in Section 2.6.2. The absorbance at 650 nm and 450 nm (representing ATPase and SOD levels, respectively) was measured using a FLUOstar Omega multi-mode microplate reader, (BMG Labtech, Ortenberg, Germany), while MH broth was used as the control.

#### 2.6.4. Effect of PV on Bacterial Cellular Metabolism

The impact of PV aqueous extracts on the cellular metabolism of MRSA (NCTC 13142) were examined using the method described by Sun et al. [28] with modification. The concentration of MRSA was adjusted to 10^8^ CFU/mL after reaching the logarithmic phase. After incubation at 37 °C for 5 h, the mixture was centrifuged at 10,000 r/min for 10 min, and the pellet was resuspended in 0.9% saline. The addition of iodonitrotetrazolium chloride (INT) to the solution (at a final concentration of 1 mmol/L) was followed by incubation at 37 °C for 30 min. The saline was used as the control. The maximum absorbance of formazan at 630 nm was then measured to assess the activity of the TCA cycle.

#### 2.6.5. Determination of Total Protein Concentration

The bacterial suspension was prepared as previously mentioned in Section 2.6.2. Subsequently, 1 mL of bacterial suspension was mixed with PV aqueous extracts to obtain a final concentration of ½ MIC and 1 MIC. After incubation for 5 h, the mixture was centrifuged at 8000 rpm for 10 min, and the supernatant was discarded. The resulting precipitate was resuspended in 1 mL PBS and homogenized in an ultrasonic ice bath (300 W, 10 min, and ultrasonic interval 1.1 s). Finally, the prepared sample was analysed for cellular protein concentration using the bicinchoninic acid (BCA) protein quantification Kit. PBS served as the control.

### 2.7. Statistical Analysis

The statistical analysis was performed using SPSS Version 20.0, and the results are presented as means ± standard deviation (SD). The mean values of multiple groups were compared using one-way analysis of variance (ANOVA), and statistical significance was determined by a *p* value of <0.05.

## 3. Results

### 3.1. Determination of Antibacterial Activity

The antibacterial assessment revealed significant effects of PV against MRSA (NCTC 13142). The ethanol and methanol extracts displayed MIC and MBC values of 5 mg/mL and 10 mg/mL, while the crude aqueous extract exhibited lower values of 2.5 mg/mL and 5 mg/mL, respectively. Consequently, the crude aqueous extract was selected for subsequent assays. The growth kinetic assay revealed a notable alteration in the growth kinetics of MRSA upon exposure to PV aqueous extract. Inclusion of the extract at 1/2 MIC and 1 MIC significantly impeded bacterial growth between 4 and 24 h when compared to the control group (*p* < 0.01; Figure 1). While both the 1/2 MIC and 1 MIC groups restricted MRSA growth, the MIC group resulted in a significantly slower growth (*p* < 0.01). These results highlighted the strong antibacterial effect of PV aqueous extract on MRSA.

In the combination testing, the PV aqueous extract yielded an FICI value of 0.75, which indicated an additive antibacterial effect in conjunction with penicillin against MRSA, and erythromycin exhibited a similar additive effect (FICI = 1) when paired with PV aqueous extract (Table 1). In addition, it was found that the use of PV aqueous extract in tandem with therapeutic antibiotics resulted in a significant decrease in antibiotic MICs. Specifically, the MIC of erythromycin was reduced by half, and that of penicillin was reduced by three-quarters, compared to their individual use. 

### 3.2. Phytochemical Compositions of PV Aqueous Extract

As shown in Table 2, 28 compounds were identified, with a matching degree of 70% with the published data. These compounds were preliminary classified into nine categories: (1) phenols/phenolic acids: protocatechuic acid, 6-gingerol, danshensu, caffeic acid, salvianolic acid A, rosmarinic acid, protocatechualdehyde, gentisic acid, anillin, isoferulic acid, and ferulic acid; (2) flavonoids: rutin and hyperoside; (3) amino acids: L-glutamic acid, L-phenylalanine and L-Tyrosine; (4) vitamin: adenine; (5) terpenoid: camphor and 18 β-Glycyrrhetintic acid; (6) nucleosides: uridine, adenosine, cordycepin and cytidine; (7) nitrogen glycosides: isoguanosine; (8) organic acids: citric acid, quinic acid and p-Hydroxy-cinnamic acid; and (9) furfural: 5-Hydroxymethylfurfural. The total ion chromatogram is presented in Figure 2.

### 3.3. Effects of PV Aqueous Extract on Cell Integrity of MRSA

At the outset, bacterial membrane damage can initiate the release of small molecules such as ions, followed by larger molecules including DNA, RNA, and proteins. To assess MRSA membrane permeability, nucleic acid and protein levels were measured before and after treatment with PV. Initial measurements showed no nucleic acid leakage, as observed by absorbance at 260 nm using UV-vis spectrophotometry. However, treatment with PV aqueous extract at 1/2 MIC and MIC for 5 h resulted in a > 6-fold increase in OD value compared to the control group (Figure 3A), indicating a leakage of bacterial nucleic acids. In addition, a significant rise in protein concentration was noted from 17.57 μg/mL to 55.92 μg/mL (*p* < 0.05) at 1/2 MIC and from 14.99 μg/mL to 133.54 μg/mL at MIC (*p* < 0.01), respectively (Figure 3B), indicating a disruption of MRSA cell membranes following the exposure of PV aqueous extracts. Furthermore, treatment with 1/2 MIC and MIC PV aqueous extract for 5 h resulted in a significant decrease in AKP activity from 0.53 U/L to 0.35 U/L and 0.30 U/L (*p* < 0.05) (Figure 3C), representing reductions of 34.47% and 41.51%, respectively. These findings suggested that PV has the potential to interfere with both the bacterial membranes and their cell walls.

### 3.4. Inhibition of PV Aqueous Extract on ATPase and SOD Activity in MRSA

The application of PV at 1/2 MIC and 1 MIC concentrations induced a marked reduction in both ATPase and SOD activity in MRSA. Specifically, treatment at 1/2 MIC and 1 MIC led to a decrease in ATPase activity from 6.27 U/mL to 4.12 U/mL and 3.40 U/mL, representing a reduction of 34.32% and 45.71%, respectively (Figure 4A). Similarly, SOD activity was reduced from 66.16% to 41.92 and 17.17% after treatment with PV at 1/2 MIC and 1 MIC (Figure 4B). These results suggest that PV may be capable of disrupting the normal cellular function of MRSA, thereby reducing both ATPase and SOD activity.

### 3.5. Effect of PV Aqueous Extract on Cell Metabolism of MRSA

The basis for analyzing cellular metabolic activity using INT is that living cells produce H^+^ during electron chain transport via the TCA cycle. This H^+^ can convert INT into a red formazan dye, which is used as a measure of bacterial metabolic activity. The treatment of PV aqueous extract at 1/2 MIC and 1 MIC for 5 h on MRSA resulted in a decrease of 24.17% and 27.17% in OD_630_ values, respectively, compared to untreated MRSA (Figure 5), suggesting the ability of PV to affect the TCA cycle in MRSA. Lower OD_630_ values indicated reduced formazan production and, therefore, a decrease in MRSA metabolic rate due to the PV extract.

### 3.6. Total Protein Content Reduction upon Treatment with PV Aqueous Extract

Analysis of the whole protein content revealed a significantly decreased MRSA protein content after treatment with PV aqueous extract compared to the control group (Figure 6). The exposure of PV aqueous extract at 1/2 MIC and 1 MIC for 5 h resulted in a reduction in total MRSA protein by 25.99% and 30.71%, respectively. The PV aqueous extract may have reduced the cellular protein content by destroying the cytomembrane and/or inhibiting its synthesis.

## 4. Discussion

The indiscriminate use of antibiotics in agricultural food production has inevitably accelerated the development of resistance in microorganisms, particularly zoonotic pathogens, resulting in a higher incidence of foodborne illnesses and greater treatment costs [29]. Compounded by the absence of new classes of therapeutic antibiotics in recent decades and a paucity of effective antibacterial agents, treating resistant bacterial infections has become a serious challenge. It is imperative to devise novel agents that can effectively control multidrug-resistant bacteria such as MRSA [30].

As is evident in the present study, aqueous, ethanol and methanol extracts of PV were able to completely suppress the growth of MRSA at 5, 10, and 10 mg/mL, respectively. The analysis of growth kinetics showed that the aqueous extract at ½ and 1 MIC levels effectively inhibited and delayed the proliferation of MRSA, providing further evidence of the antibacterial efficacy of PV. The antibacterial potential of different PV extracts against foodborne pathogens has been previously documented [31]. It is worth noting that the choice of extraction solvents can selectively isolate specific classes of phytochemicals from plant materials, consequently impacting the biological activity of the resulting plant products [32,33]. Although ethanolic extracts have been reported to exhibit greater antibacterial activity than their aqueous counterparts, probably due to the extraction of more polyphenols from plants [34], the present study revealed that the PV aqueous extract displayed greater effectiveness against MRSA in comparison to 100% methanolic and 70% ethanolic extracts. This could be attributed to the use of ultrasonication in our study, which likely facilitated the release of a larger quantity of phytochemicals from the plant cells. Notably, a higher total phenolic content (TPC) was found in the PV aqueous extract compared to the ethanol and methanol extracts [17]. Moreover, a strong correlation was found between TPC levels and *S. aureus* growth suppression in tests with berry extracts [35]. Nevertheless, our results may suggest that the anti-MRSA activity of PV may stem from the collective effects of multifunctional component groups present in the aqueous extract.

PV is commonly known as a self-healing plant and contains several classes of phytochemicals, such as triterpenoids, phenolics, flavonoids, and tannins, as the main bioactive components [36]. In our study, HPLC-ESI-MS/MS tentatively identified 28 compounds present in the PV aqueous extract, including various phenolic acids and flavonoids previously reported in the literature, as well as terpenoids and organic acids [37]. Notably, several of these compounds, such as phenolic acids (e.g., rosmarinic acid, protocatechuic acid, and caffeic acid), flavonoids (e.g., rutin), and furfural have been shown to possess antibacterial properties [38,39,40,41]. The intricate blend of multiple components within the crude aqueous extract raises the possibility that its biological impact may diverge greatly from that of a single component group.

Combination therapy has emerged as a proven and effective treatment strategy in the management of resistant bacterial infections [42]. A growing body of evidence points to the potential for synergistic effects when plant-derived phytochemicals are combined with clinical antibiotics. Our study highlights the potential of the PV aqueous extract to augment the efficacy of penicillin or erythromycin against MRSA. A previous study demonstrated a remarkable synergy between *Daphne genkwa* extract and oxacillin in inhibiting MRSA [43]. Rosmarinic acid, one of the main compounds in the PV extracts, has been found to exhibit synergistic effect with vancomycin against *S. aureus* and MRSA [39]. These results imply that the active constituents of these extracts may be able to overcome the resistance of MRSA. Although the PV extract shows clear anti-MRSA effects, the precise nature of its mode of action remains unknown. We therefore undertook a comprehensive assessment of various cellular functions, including membrane permeability, intracellular enzymatic activity, and metabolism, following treatment with the crude aqueous extract.

At the core of cell biology, cell membranes serve as a selective osmotic barrier, shielding cells from harmful substances while permitting the passage of essential nutrients required for survival [44,45]. Therefore, a leakage of the cell constituent can act as a crucial indicator of cell membrane integrity [46]. In fact, the leakage of large molecules like nucleic acids (e.g., DNA and RNA) and proteins from the cell is often an important sign of compromised cell membrane function. Our study revealed a clear and pronounced leakage of nucleic acid and protein from MRSA, as evidenced by a substantial increase in absorbance at 260 nm and protein concentration following treatment with PV aqueous extracts. In addition, our study uncovered the intriguing observation that PV is capable of inflicting membrane damage even at low concentrations (i.e., ½ MIC). As the concentration approached the MIC, the extent of the damage became considerably pronounced, leading to a significant 10-fold increase in extracellular protein concentration.

AKP is an enzyme that plays a crucial role in bacterial cell wall metabolism and is involved in the synthesis of peptidoglycan, a major component of Staphylococcus. Therefore, its involvement in these processes makes AKP crucial to the function and maintenance of the bacterial cell walls. Our investigation reveals a noteworthy reduction in AKP activity in MRSA, following a 5 h PV treatment, with a concomitant decrease in intracellular AKP levels. This observation concurs with previous findings of reduced AKP activity in *L. monocytogenes* post treatment with clove oil [27]. Furthermore, the deactivation of AKP hinders cell differentiation, alters the bacterial cell wall, and modifies its properties, potentially influencing bacterial growth and virulence.

Cell wall and membrane disruption, and the leakage of nucleic acids and proteins, may alter cell membrane energy transduction systems, thus leading to further damage and finally the death of cells. Notably in our study, the treatment of MRSA with PV aqueous extracts at concentrations of ½ MIC and 1 MIC yielded a significant reduction in both ATPase and SOD activity. Lower ATPase levels can trigger the efflux of ATP from bacterial cells, ultimately impeding their cellular respiratory metabolism. Research has shown that the application of pulsed light treatment significantly decreased the ATP content of *E. coli* O157:H7, thereby impacting its vitality [47]. Cui et al. [48] found that the application of cinnamon essential oil suppressed cellular respiration in both *S. aureus* and *E. coli*, as evidenced by alterations in ATP concentration. SOD is a pivotal enzyme that safeguards cells from oxidative stress by neutralizing oxidized free radicals, thus playing a vital role in the cellular defence against reactive oxygen species (ROS) [49]. The results of our study strongly suggested that PV extract may serve as an effective inhibitor of bacterial growth by obstructing the transmission of both information and energy.

The TCA cycle functions as the primary metabolic process for generating energy through the complete oxidation of nutrients. The present study illustrated that treatment with PV aqueous extract at 1/2 MIC and 1 MIC levels induces a decline in TCA cycle activity in MRSA. A weakening TCA cycle can lead to decreased cellular respiration, resulting in low energy production and, eventually, bacterial death. Observing the impact of ε-Polylysine on *Shewanella putrefaciens* revealed that an increasing amount of plant extract can progressively reduce and eventually block the pathogen’s TCA cycle [50]. Villegas-Mendoza et al. utilized formazan yield to determine the in vivo cellular respiration rate of marine bacteria, concluding that impaired cellular respiration and insufficient energy supply could result in subsequent bacterial death [51].

Furthermore, our investigation indicated that treatment with PV aqueous extracts caused a substantial decrease in the total protein content of MRSA, potentially due to the disruptive influence of diverse bioactive components present within the plant extract on the MRSA cell wall and membrane. This may subsequently lead to attenuated protein synthesis, as previous studies have illustrated that ursolic acid initially impaired MRSA membrane integrity, consequently leading to the inhibition of protein synthesis [52]. Importantly, it has been documented that PV extract possesses the ability to impede protein biosynthesis and proficiently obliterate bacterial cells [53].

## 5. Conclusions

This comprehensive investigation has revealed the multifaceted functionalities of the traditional edible medicinal plant PV. The study provides robust evidence of its remarkable proficiency in disrupting the proliferation, metabolism, and survival mechanisms of the bacterial pathogen MRSA. The validation of the antibacterial mode of action, along with its ability to enhance the efficacy of clinical antibiotics, highlights its potential as an alternative and complementary anti-MRSA phytotherapeutic agent. Furthermore, its role in ensuring food safety by controlling pathogenic bacteria is underscored. Future research may explore the health benefits and functional food attributes of *P. vulgari* given the escalating interest in the use of natural compounds over synthetic agents, and the growing trend of plant-based diets. Such inquires have the potential to fully unlock the incredible potential of this traditional medicinal plant.

## Figures and Tables

**Figure 1 foods-13-00660-f001:**
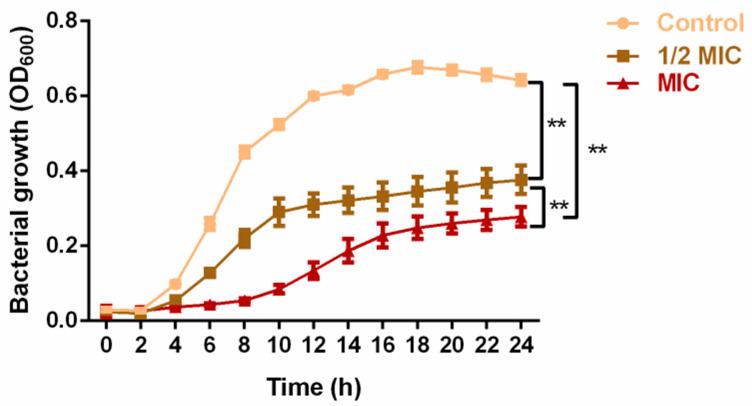
Inhibitory effect of PV aqueous extract at 1/2 MIC and MIC concentrations on MRSA growth over a period of 24 h. Values are represented as mean ± SD of three independent experiments. (** *p* < 0.01 vs. control).

**Figure 2 foods-13-00660-f002:**
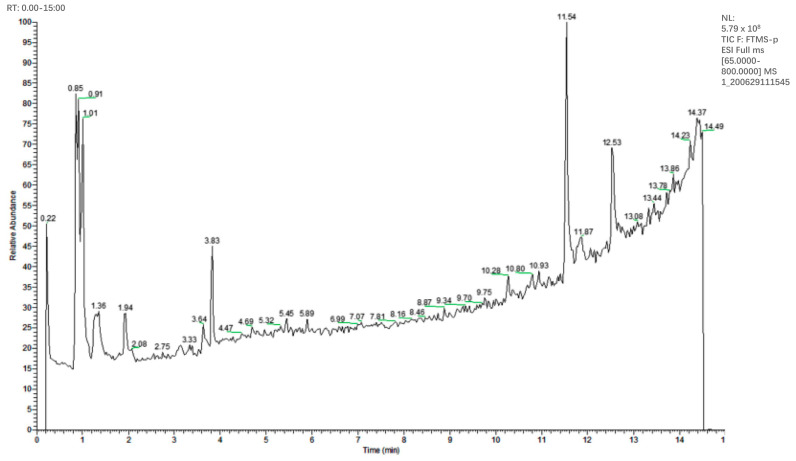
HPLC-MS negative ion chromatogram of the PV extract.

**Figure 3 foods-13-00660-f003:**
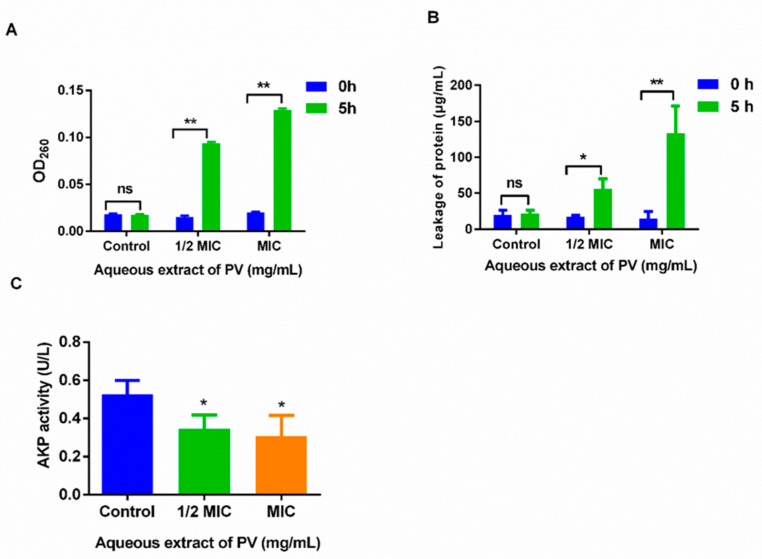
The effect of the cell membrane permeability and cell wall permeability of MRSA after treatment with PV aqueous extract. (**A**) The release of 260 nm absorbing materials of MRSA after treatment with PV extract. (**B**) Protein leakage from MRSA treated with PV aqueous extract at 1/2 MIC and MIC. (**C**) AKP activity after treatment with PV aqueous extract. Values are expressed as mean ± SD of three independent experiments. (* *p* < 0.05, ** *p* < 0.01 vs. Control; ns: not significant).

**Figure 4 foods-13-00660-f004:**
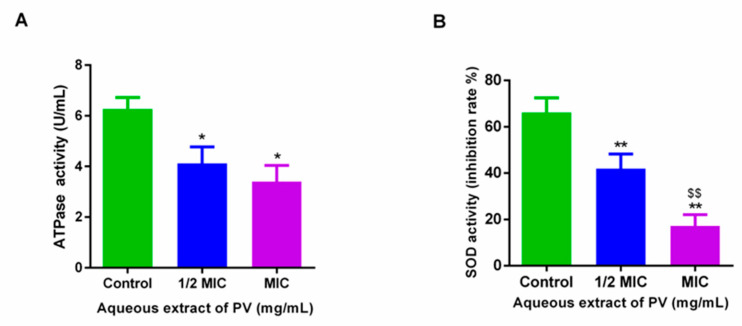
Effects on ATPase (**A**) and SOD (**B**) enzymes in MRSA treated with PV aqueous extract. Results are shown as mean ± SD (* *p* < 0.05, ** *p* < 0.01 vs. control, ^$$^ *p* < 0.01, 1/2 MIC group vs. MIC group).

**Figure 5 foods-13-00660-f005:**
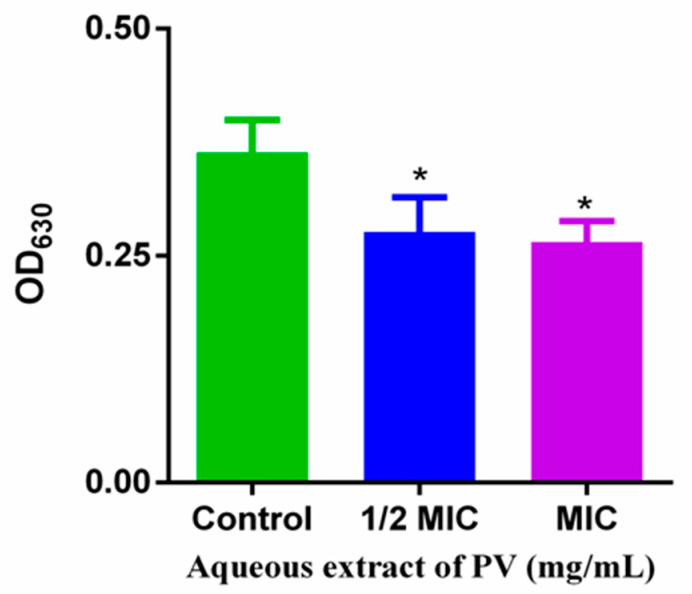
Effects of PV aqueous extracts on MRSA metabolism. Values are means of three independent experiments, expressed as mean ± SD (* *p* < 0.05 vs. control).

**Figure 6 foods-13-00660-f006:**
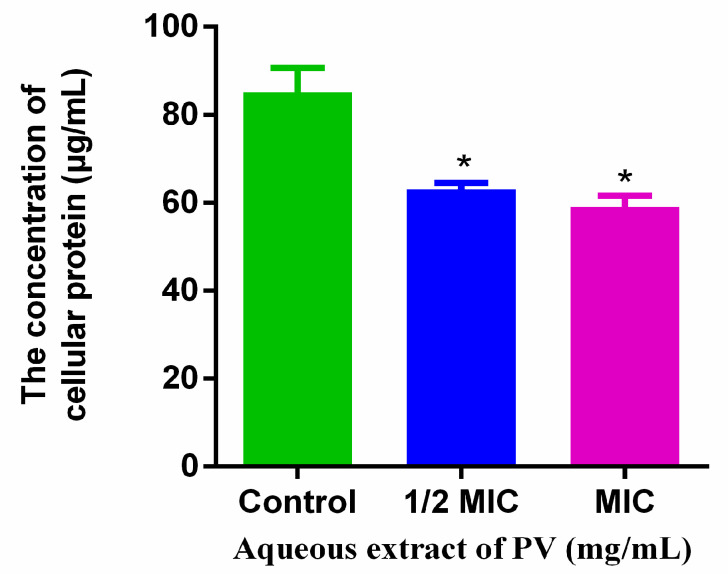
Effects of PV extracts on the total cellular protein. Values are expressed as mean ± SD of three independent experiments. (* *p* < 0.05 vs. control).

**Table 1 foods-13-00660-t001:** FICI of PV extract for MRSA NCTC 13142 in combination with penicillin or erythromycin.

Antibiotics	Antibiotic Alone MIC (mg/L)	PV Extract Alone MIC (mg/mL)	Combination Antibiotic MIC (mg/L)	Combination PV Extract MIC (mg/mL)	FICI	Outcome
Penicillin	64	2.5	16	1.25	0.75	additivity
Erythromycin	128	2.5	64	1.25	1	additivity

**Table 2 foods-13-00660-t002:** Table identification of compounds in PV aqueous extract by UPLC-ESI-MS/MS.

Number	RT (min)	Molecular Weight	Type	Elemental Composition	Tentative Identification	Intensity
1	0.48	126.03	[M − H]^−^	C_6_H_6_O_3_	5-Hydroxymethylfurfural	200,786,294.28
2	0.85	192.03	[M − H]^−^	C_6_H_8_O_7_	Citric acid	231,224,124.95
3	0.91	147.05	[M − H]^−^	C_5_H_9_NO_4_	L-Glutamic acid	118,909,823.16
4	0.93	192.06	[M − H]^−^	C_7_H_12_O_6_	Quinic acid	121,100,618.62
5	1.08	154.03	[M − H]^−^	C_7_H_6_O_4_	Protocatechuic acid	29,413,149.12
6	1.10	294.18	[M − H]^−^	C_17_H_26_O_4_	6-Gingerol	117,982,186.76
7	1.35	198.05	[M − H]^−^	C_9_ H_10_O_5_	Danshensu	82,420,769.72
8	1.93	180.04	[M − H]^−^	C_9_H_8_O_4_	Caffeic acid	284,978,181.05
9	2.07	244.07	[M − H]^−^	C_9_H_12_N_2_O_6_	Uridine	10,320,534.18
10	2.54	164.05	[M − H]^−^	C_9_H_8_O_3_	p-Hydroxy-cinnamic acid	20,103,597.17
11	2.63	283.09	[M − H]^−^	C_10_H_13_N_5_O_5_	Isoguanosine	26,317,693.32
12	2.76	135.05	[M − H]^−^	C_5_H_5_N_5_	Adenine	28,285,891.13
13	2.84	165.08	[M − H]^−^	C_9_H_11_NO_2_	L-Phenylalanine	60,424,033.14
14	3.15	267.10	[M − H]^−^	C_10_H_1_N_5_O_4_	Adenosine	150,919,822.07
15	3.26	251.10	[M − H]^−^	C_10_H_13_N_5_O_3_	Cordycepin	15,636,305.51
16	3.33	494.12	[M − H]^−^	C_26_H_22_O_10_	Salvianolic acid A	9,241,354.22
17	3.82	360.08	[M − H]^−^	C_18_H_16_O_8_	Rosmarinic acid	466,272,838.52
18	3.95	138.03	[M − H]^−^	C_7_H_6_O_3_	Protocatechualdehyde	201,195,637.62
19	4.68	610.15	[M − H]^−^	C_27_H_30_O_16_	Rutin	23,523,415.72
20	4.69	464.10	[M − H]^−^	C_21_H_20_O_12_	Hyperoside	34,180,910.77
21	14.75	152.12	[M − H]^−^	C_10_H_16_O	Camphor	130,897,410.82
22	1.08	154.03	[M − H]^−^	C_7_ H_6_O_4_	Gentisic acid	29,413,149.12
23	3.95	152.05	[M − H]^−^	C_8_ H_8_O_3_	Anillin	18,295,473.19
24	1.74	181.07	[M − H]^−^	C_9_H_11_NO_3_	L-Tyrosine	4,548,704.97
25	3.82	194.06	[M − H]^−^	C_10_H_10_O_4_	Isoferulic acid	82,584,298.91
26	5.44	470.34	[M − H]^−^	C_30_H_46_O_4_	18 β-Glycyrrhetintic acid	6,110,146.94
27	1.66	243.09	[M − H]^−^	C_9_H_13_N_3_O_5_	Cytidine	14,310,456.88
28	3.82	194.06	[M − H]^−^	C_10_H_10_O_4_	Ferulic acid	82,584,298.91

## Data Availability

The original contributions presented in the study are included in the article, further inquiries can be directed to the corresponding authors.

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
