# Peer review of "Exploring the Antibacterial Potential and Underlying Mechanisms of Prunella vulgaris L. on Methicillin-Resistant Staphylococcus aureus"

_foods, 2024, doi:10.3390/foods13050660_

Round 1
Reviewer 1 Report
Comments and Suggestions for Authors
The manuscript from Li and collaborators describes the antibacterial potential of crude extracts from Prunella vulgaris. L against Staphylococcus aureus and attempts to provide insights into some subcellular effects involved in this action.
Although the manuscript is well written and in general well presented, it is too simple for Foods.
Major issues:
- The authors only have used crude extracts in this study. Even though they have primarily characterized the most active extract, the antimicrobial and chemical composition of this species have been reported before by other groups.
- The reported activity of the most active extract is way too high: 2,5 mg/mL. Such high concentration is not considered a relevant result.
- The assays employed a single MRSA strain. It would be important to evaluate the effects using strains with different phenotypes.
- The assays employed to evaluate the “antimicrobial mechanism” are too preliminary. The authors should include electronic microscopy and other methods more appropriate for evaluation of the mechanisms (flow cytometry, fluorescence microscopy, etc).
- Some results from “mechanisms assays” should be better interpreted. For instance, in the measurement of AKP activity, is such small reduction relevant (even though the values are statistically different)?
- Given the high concentrations needed for MRSA inhibition, the toxicity of the extract should be evaluated.
Other minor issues are included in the attached pdf file

Author Response
Dear Reviewer,
We appreciate your valuable feedback. We have revised and incorporated additional information into the manuscript (text in red) during the revision process. We believe that these improvements will elevate the manuscript to meet the standards of the Foods journal.
Please find our responses to each of your comments in the document attached
.

Reviewer 2 Report
Comments and Suggestions for Authors
My dear colleagues,
You really did a good job. My congratulations. I must admit I really enjoyed in your manuscript but I have some doubts about the possibility of using aqueous extract of Prunella vulgaris L. in food industry, or precisely in food treatment against foodborne pathogens. Namely, I really wonder:
1. Does the food treatment with Prunella vulgaris L. aqueous extract changing the sensoric characteristics of food?
2. If yes, how? (for instance, improving by making the food more spicy or making it worse)
Yes, I know that wasn't the aim of your research, I really know what you are thinking right now, but it will good to consider my questions, for planning the further research. Namely, if the treatment has a negative impact on food sensoric value, than you must limit it only to the treatment of tools and work surfaces in the production process. Please, consider my curiosity if you will design the further research which might be the logical sequel of this one.
However, I kindly ask you to edit your manuscript . For instance:
- The chapter 2. Materials and methods is positioned on the bottom of the page 2. (?!)
- The title of Fig. 2 "Figure 2. HPLC-MS total ion chromatogram of the PV extract." is placed (it "ran away") on page 8, but the whole chromatogram is placed on the bottom on previous page.
- Please delete all the points on the end of titles of all tables and figure. Why is that, anyway?!
- Also, the chapter 5. Conclusion is placed on the bottom of page 12, but the whole text is on next page.
- However, according to writing rules of botanical systematics, the initial of taxonomist must be written in normal font, but not in italic. Only the latin names of plant species must be written in italic, but initial of Carl Linnaeus surname must be in normal font.
- Also, you started the first sentence of abstract by Prunella vulgaris Linn instead of Prunella vulgaris L.
Please correct this in final version. Thank you!
Author Response
Dear Reviewer,
Thank you for your positive and encouraging remarks. We have addressed each of your comments (text in red) in the attached document.

Reviewer 3 Report
Comments and Suggestions for Authors
Author Response
Dear Reviewer,
Thank you for providing comments highlighted in the text. As there are no specific questions associated with the highlighted text, we are unsure about how to address them. Please find our responses to each highlight in the attached document.

Reviewer 4 Report
Comments and Suggestions for Authors
Exploring the antibacterial potential and underlying mechanisms of Prunella vulgaris. L on methicillin-resistant Staphylococcus aureus
The manuscript deals with a subject that is not new (Groşan, Alexandra & Vari, Camil-Eugen & Ștefănescu, Ruxandra & Danciu, Corina & Pavel, Ioana & Dehelean, Cristina & Man, Adrian & David, Remona & Vlase, Laurian & Muntean, Lucia. (2020). Antibacterial and antitumor activity of the species Prunella vulgaris L.. Revista Romana de Medicina de Laborator. 28. 405-417. 10.2478/rrlm-2020-0031, etc...), but for the journal it could be; it has interesting results, but the writing of this paper need be improved. However, there are some errors (in my opinion) in the manuscript and in present form several explanations and modifications are needed. In general, the article was well written and presents a topic relevant to academia. I take this opportunity to congratulate the authors for their work.
I have written some suggestions as a way to further improve the study.
Multidrug resistance among Staphylococcus aureus is a serious public health threat, necessitating the development of new antimicrobial drugs from either synthetic or natural sources. In this study, the authors Ziyin Li et al. report on the antibacterial potential of Prunella vulgaris L. against MRSA and its mechanism. They identify phytochemical components in the aqueous extract of PV. Complementary analyzes and determinations support the authors' theory.
Specific comments are provided below:
Abstract
Line 18: Staphylococcus aureus instead of staphylococcus aureus
Line 19-20: PV contains active substances (also specified in the article at line 362) that are soluble in alcohol, and the extraction of these substances can be more efficient using ethaol. The ethanol can extract compounds such as flavonoids, tannins, triterpenoids and phenolics as well as essential oils, which may have antibacterial properties. Extraction using water can be effective to obtain substances such as polysaccharides, alkaloids or other components with antibacterial potential. Indeed, the sensitivity of MRSA to herbal actives can vary, and there is no one-size-fits-all approach. However, specific research on the antibacterial activity of most plants and their extracts shows that alcoholic extracts are “stronger” than aqueous ones.
Please explain this aspect, becouse at line 19-20 (in Abstract) you mention: “The aqueous extract of PV demonstrated greater anti-MRSA activity compared to the ethanolic and methanolic extracts.”
Line 47: Staphylococcus aureus instead of S. aureus. Is the first appearance in the text.
Line 82: Escherichia coli instead of E. coli and Klebsiella pneumoniae instead of K. pneumonia
Materials and Methods
Line 174: penicillin and erythromycin - where were purchased from and what was the initial concentration?
Line 185: “bacteria were cultured” – in what culture medium?
Line 189: “the leakage of the enterocyte”??? Here I did not understand what you wanted to specify!!!!! Enterocytes were not used, but bacterial cells from which components were released following membrane rupture.
Line 197: “using a standard curve” - how was the standard curve obtained?
Line 200: “Bacteria grown to the logarithmic phase” – in what culture medium?
Line 220: “final concentrations equivalent to 0, 0.5 and 1 MIC” - how do you mean concentrations equivalent to 0 MIC?
Results
Figure 1, 2, 3, 4, 5, 6: what was used for the control? It must be specified in Material and methods for each method.
Table 1: what does mean “Combination Antibiotic” and “Combination PV extract”? In Section 2.5.3. it said that PV extract was combined with either penicillin or erythromycin. Please explain or reformulate in the text for a better understanding!
Line 441: I do not think it is appropriate to present figure no. 7. I would eliminate it!
Conclusion
Line 443-453: The conclusions seem very ... general. I kindly request the authors to reformulate them and point out also the evidence, arguments or premises from the research paper that led to the hypothesis according to which the Prunella vulgaris L. has an antibacterial effect on MRSA.
I have no other objections.
Comments on the Quality of English LanguageI detected only small language mistakes.
Author Response
Dear Reviewer,
We appreciate your valuable feedback and the opportunity to refined our work. We will make the necessary revision to ensure that our manuscript aligns more closely with the scholarly expectations of the journal.
Please see attached our responses to your comments.

Round 2
Reviewer 1 Report
Comments and Suggestions for Authors
The authors have not provided sufficient information for change my first opinion on this manuscript
Author Response
Dear Reviewer,
Your insightful comments are greatly appreciated and valued. We have included additional information to some of your original comments from the initial revision (please see attached response). We believe that we have comprehensively addressed all your concerns.
Best regards
Authors

Reviewer 3 Report
Comments and Suggestions for Authors
I have no further comments to make to the authors. I am satisfied with the changes they made according to my suggestions.
Author Response
Dear Reviewer,
We express gratitude for approving our revised manuscript. Your comments and suggestions are highly valued.
With Best Regards
Authors
Reviewer 4 Report
Comments and Suggestions for Authors
The author has made revisions to address all the issues pointed out in the first review. The revised paper has a rigorous logic and an accurate narration. The article is well written, although it does not have a vast complexity. Anyway, I would like to congratulate the authors for the improved revised manuscript.
I no longer have any clarification related to the content of the article, but only related to the last figure. If you decided to keep figure 7 (which seems too pedagogical / didactic to me), I suggest you insert it in the text. A chapter should not end with a figure, but with its description.
Author Response
Dear Reviewer,
We express gratitude for approving our revised manuscript. Your insightful comments and suggestions are highly valued.
We have deleted Fig 7 as suggested in the final revision.
With Best Regards
Authors